

# Machine learning driven by environmental covariates to estimate high-resolution PM2.5 in data-poor regions

XiaoYe Jin[1,2], Jianli Ding[1,2,3], Xiangyu Ge[1,2], Jie Liu[1,2], Boqiang Xie[1,2], Shuang Zhao[1,2] and Qiaozhen Zhao[1,2]

[1] Department of MOE Key Laboratory of Oasis Ecology, Xinjiang University, Urumqi, China
[2] College of Resources and Environment Science, Xinjiang University, Urumqi, China
[3] MNR Technology Innovation Center for Central Asia Geo-Information Exploitation and Utilization, Urumqi, China

## ABSTRACT

$PM_{2.5}$, which refers to fine particles with an equivalent aerodynamic diameter of less than or equal to 2.5 $\mu$m, can not only affect air quality but also endanger public health. Nevertheless, the spatial distribution of $PM_{2.5}$ is not well understood in data-poor regions where monitoring stations are scarce. Therefore, we constructed a random forest (RF) model and a bagging algorithm model based on ground-monitored $PM_{2.5}$ data, aerosol optical depth (AOD) and meteorological data, and auxiliary geographical variables to accurately estimate the spatial distribution of $PM_{2.5}$ concentrations in Xinjiang during 2015–2020 at a resolution of 1 km. Through 10-fold cross-validation (CV), the RF model and bagging algorithm model were verified and compared. The results showed the following: (1) The RF model achieved better model performance and thus can be used to estimate the $PM_{2.5}$ concentration at a relatively high resolution. (2) The $PM_{2.5}$ concentrations were high in southern Xinjiang and low in northern Xinjiang. The high values were concentrated mainly in the Tarim Basin, while most areas of northern Xinjiang maintained low $PM_{2.5}$ levels year-round. (3) The $PM_{2.5}$ values in Xinjiang showed significant seasonality, with the seasonally averaged concentrations decreasing as follows: winter (71.95 $\mu$g m$^{-3}$) > spring (64.76 $\mu$g m$^{-3}$) > autumn (46.01 $\mu$g m$^{-3}$) > summer (43.40 $\mu$g m$^{-3}$). Our model provides a way to monitor air quality in data-scarce places, thereby advancing efforts to achieve sustainable development in the future.

Corresponding author
Jianli Ding, dingjl@163.com,
watarid@xju.edu.cn

## INTRODUCTION

$PM_{2.5}$, which refers to fine particles with an equivalent aerodynamic diameter of 2.5 $\mu$m or less, is the main cause of air pollution (*Goldberg et al., 2019*; *Nel, 2005*; *Zhang et al., 2017a*). Because $PM_{2.5}$ threatens urban ecosystems and human health, the management of air pollution is crucial to achieving and advancing sustainable development. High $PM_{2.5}$ concentrations not only directly cause respiratory diseases but also weaken the gastrointestinal, cardiovascular, and immune systems (*Leiva et al., 2013*; *Muhlfeld et al.,*

*2007*; *Nabavi, Haimberger & Abbasi, 2019*). Exposure to high levels of PM$_{2.5}$ can also result in cancer and even death (*Eilstein, 2009*). For instance, air pollution was deemed responsible for 1.24 million deaths in China during 2017, 851,600 of which were caused by atmospheric PM$_{2.5}$ (*Yin et al., 2020*). Consequently, PM$_{2.5}$ jeopardizes the United Nations' Sustainable Development Goals. Furthermore, PM$_{2.5}$ is suspended in the atmosphere for an extended period of time and can be carried to neighboring places by atmospheric circulations, resulting in regional air pollution (*Gautam et al., 2016*). In particular, PM$_{2.5}$ pollution poses a considerable hazard to public health and the environment in the areas surrounding rapid urbanization in China. Therefore, it is of great significance to monitor PM$_{2.5}$ for urban sustainable development and human well-being in a timely and accurate manner.

Traditionally, PM$_{2.5}$ concentrations at observation points have been routinely sampled in real time at ground monitoring stations, with the resulting measurements being highly precise and continuous in time. Despite these benefits, these monitoring stations are deployed with low density, making it impossible to accurately determine the distribution of PM$_{2.5}$. Specifically, the station density is high in economically developed regions, but the density is low in less developed areas; in other words, monitoring stations are sparse and are focused mostly in discrete cities. Nevertheless, with the rapid advancement of remote sensing technologies, the gaps in the monitoring station distribution can be filled (*Pu & Yoo, 2021*; *Sun, Gong & Zhou, 2021*). Satellite remote sensing provides an efficient method to rapidly and economically predict PM$_{2.5}$ concentrations by using aerosol optical depth (AOD) estimates in the areas devoid of monitoring stations, enabling the acquisition of surface PM$_{2.5}$ data on a large scale.

Previous studies on the relationship between PM$_{2.5}$ and AOD mostly adopted linear or multiple regression models, with the linear relationship between AOD and PM$_{2.5}$ being the main focus. For instance, Engel-Cox, Holloman et al. (*Engel-Cox et al., 2004*) discovered a linear association between AOD and PM$_{2.5}$ with a correlation coefficient of 0.4. In contrast, physical and chemical models are slower and have more complicated features than linear or multiple regression models (*Lin et al., 2015*; *Yang, Xu & Jin, 2019*; *Zhang & Li, 2015*). Nevertheless, it is challenging for statistical models to account for the impacts of geographical and temporal variations on the estimation. Moreover, while geostationary methods can effectively solve the problems making it difficult to ascertain the spatial distribution of PM$_{2.5}$, spatial differences and other methods demand that some prerequisite conditions (*i.e.*, the number and distribution of stations) be satisfied. In the last decade, the extensive use of machine learning has made it possible to accurately estimate the spatial and temporal distributions of PM$_{2.5}$. Examples of such algorithms include the random forest (RF) model, the geographically weighted regression (GWR) model, hierarchical models, and Bayesian models (*Song et al., 2015*; *Zhai et al., 2018*). Because of its superior ability to choose and use various independent parameters that may affect the prediction of dependent variables, machine learning can better predict the concentrations of air pollutants than can the other techniques mentioned above (*Yang et al., 2022*). For instance, to demonstrate the high predictability of PM$_{2.5}$ concentrations in the Beijing–Tianjin–Hebei region of China, *Zhao et al. (2020)* used an RF model to

add auxiliary variables, and the coefficient of determination ($R^2$) reached 0.86. On the other hand, machine learning algorithms need to be driven by a large number of samples. Moreover, even though $PM_{2.5}$ ground stations are sparsely distributed, observations are acquired every minute, and thus, the volume of temporally continuous data compensates for the spatial heterogeneity of stations.

Nevertheless, various meteorological factors, geographical factors and the resolution of AOD can affect the accuracy of the model. Thus, incorporating multiple factors can better reflect the correlation between AOD and $PM_{2.5}$. For example, *Hu et al. (2017)* applied an RF model using AOD and 39 auxiliary factors to estimate the $PM_{2.5}$ concentrations in the United States and obtained an overall $R^2$ of 0.80 with a root mean square error (RMSE) of 2.83 $\mu g\ m^{-3}$. Furthermore, *Hu et al. (2013)* developed a GWR model that included AOD, meteorological factors, and land use/cover change (LUCC), yielding an $R^2$ of 0.82. Subsequently, *Brokamp et al. (2018)* utilized AOD and 11 parameters with an RF model to estimate the $PM_{2.5}$ concentrations within an area containing seven counties in the United States, and the estimated $PM_{2.5}$ exhibited relatively strong agreement with the $PM_{2.5}$ observations ($R^2 = 0.92$). MOD04_3K, MOD04_L2, MYD04_3K, and MYD04_L2 were commonly utilized in early studies (*Sahu et al., 2020*; *Xu, Huang & Guo, 2021b*). Given the release of the MCD19A2 product in 2018, it presently possible to refine the estimates of $PM_{2.5}$. This product utilizes image-based processing in conjunction with time series analysis (*Lyapustin et al., 2011*; *Pu & Yoo, 2021*). As a result, aerosol inversion and atmospheric correction can be performed on sparsely vegetated land and relatively bright surfaces. This product has a high spatial resolution of 1 km, and high-quality AOD data at this resolution with great performance can better explain the aerosol distribution than similar data at a coarse spatial resolution. Therefore, the MCD19A2 product has been favored by many researchers (*Goldberg et al., 2019*; *Jia et al., 2020*; *Nabavi, Haimberger & Abbasi, 2019*; *Pu & Yoo, 2021*).

This study has three objectives: (1) to establish a Xinjiang-specific RF model and bagging algorithm model incorporating AOD, meteorological variables, and other auxiliary data and to evaluate the performance of both models; (2) to understand the long-term spatial and temporal distribution changes in $PM_{2.5}$ throughout Xinjiang from 2015 to 2020; and (3) to study the relationships between the annual and seasonal variations in AOD and $PM_{2.5}$ in Xinjiang. After obtaining high-resolution (1-km) AOD data and combining them with auxiliary data comprising meteorological and geographical variables, we used a model to estimate the $PM_{2.5}$ concentrations across Xinjiang during 2015–2020. The research findings are useful not only as a reference for the future satellite retrievals of $PM_{2.5}$ concentrations but also as a tool for estimating $PM_{2.5}$ exposure.

## DATA AND METHODS

### Study area

Xinjiang is located in northwest China (73°40′E∼96°18′E, 34° 25′N∼48°) and is China's largest autonomous region, accounting for one-sixth of China's total area. Far from the ocean, Xinjiang is landlocked far inland and exhibits a typical temperate continental

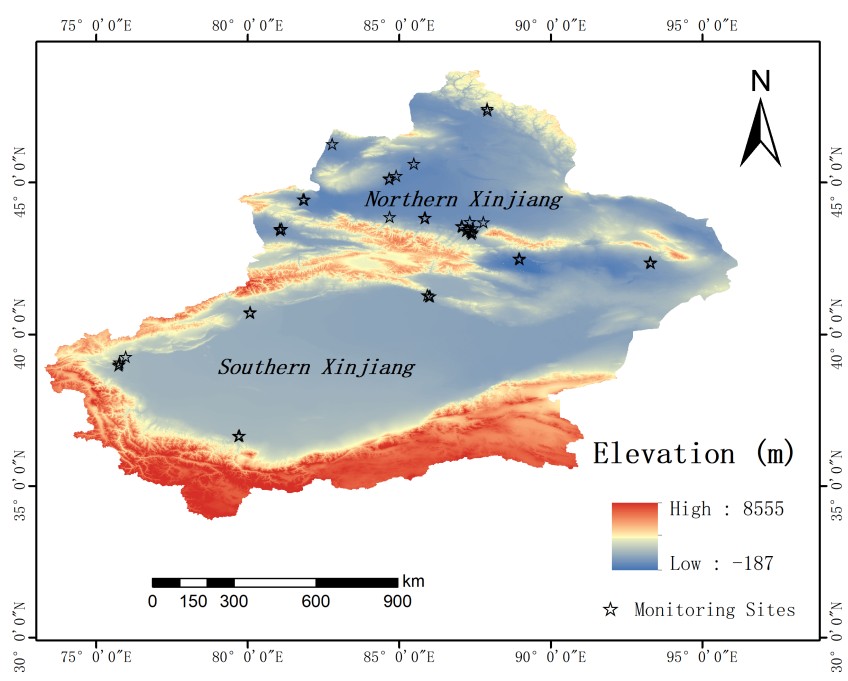

**Figure 1** **Map of Xinjiang with the location of monitoring sites.** The stars indicate monitoring points.

climate characterized by cold winters and hot summers, a dry climate, a large temperature difference between day and night, low precipitation, and annual precipitation mostly below 150 mm (*Gao et al., 2011*). Xinjiang has a special economic strategic position: it not only shares borders with many other countries but also represents the forefront of China's expansion into Central Asia and constitutes the current core region of China's Belt and Road Initiative (*Liu & Cao, 2013*). However, Xinjiang is also one of China's main sources of atmospheric dust, which has a significant effect on local air quality (*Liu et al., 2021*; *Mao et al., 2005*). Forty-one air quality monitoring stations are deployed across Xinjiang, and their locations are plotted in Fig. 1.

## Data
### MODIS AOD data

We gathered AOD information throughout Xinjiang from January 2015 to February 2021 utilizing the Moderate Resolution Imaging Spectroradiometer (MODIS) instruments carried by the Terra and Aqua satellites. Terra and Aqua provide daily observations of the global equator at local crossing times of 10:30 am and 1:30 pm, respectively. Based on a new advanced algorithm, the atmospheric correction is realized from multiple angles. By analyzing time series images of bright and dark vegetation surfaces, AOD estimates over both types of vegetation surfaces can be accurately retrieved with a resolution of 1 km (*Zhang et al., 2019b*). The Terra and Aqua combined aerosol product MCD19A2 (550 nm, downloaded from https://aeronet.gsfc.nasa.gov) consists of various data layers. Our research is based on the terrestrial 550 nm (green band) AOD.

### Ground-level PM$_{2.5}$ monitoring

The PM$_{2.5}$ ground measurement data we collected from January 2015 to February 2021 were downloaded from China's National Air Quality Real-time Release Platform (http://106.37.208.233:20035/). The platform has published hourly PM$_{2.5}$ concentrations from air quality monitoring stations in more than 1,600 major Chinese cities since 2013. In the main analysis, we limited the study area to the Xinjiang region of China. Xinjiang contains 41 ground air quality monitoring stations in 16 cities spread across an area exceeding 1.6 million km$^2$. The locations of the research area and the ground monitoring stations are shown in Fig. 1. The hourly PM$_{2.5}$ concentrations on this platform are measured and reported in accordance with China's National Ambient Air Quality Standard, and the accuracy and quality control of these PM$_{2.5}$ measurements have been previously reported (*Chen et al., 2018*). The ground PM$_{2.5}$ monitoring information is shown in Table 1.

### Meteorological data

The meteorological data were obtain from the second edition of the National Centers for Environmental Prediction (NCEP) climate forecasting system, which provides meteorological data every 6 h. Fifteen meteorological variables were considered herein: the dew point temperature at 2 m, maximum/minimum temperature at 2 m, maximum/minimum humidity, land surface temperature (LST), relative humidity (RH), precipitation, potential evapotranspiration rate, upward/downward longwave radiation (ULR/DLR), upward/downward shortwave radiation (USR/DSR), surface pressure (SP), and wind speed (WS). The above data can be downloaded from Google Earth Engine (https://earthengine.google.com/).

### Auxiliary data

The land use parameters employed herein include the normalized difference vegetation index (NDVI), enhanced vegetation index (EVI), MODIS NDVI (product name: MYD09GA) and MODIS EVI (product name: MOD09GA) provided by NASA. The predictive factors of the model include the population density and the drought index (DI), namely, the Keetch-Byram drought index (KBDI), which is estimated from the ground temperature and precipitation at a weather station and is manually interpolated and refined by experts. The above data can be downloaded from Google Earth Engine (https://earthengine.google.com/).

## Model development
### Bagging algorithm

The bagging algorithm is a popular, simple and effective ensemble learning algorithm. Proposed by Breiman in 1996, the bagging algorithm can be used for both classification and regression. The algorithm selects from the original data set with replacement; each sample is independent of each other, each subset in the training set is used to train the classifier, and finally, the results of each classifier are combined by voting. The model is guaranteed to achieve high performance, as is the statistically reliable estimation arising from the generalization ability of the model, and the risk of overfitting is avoided.

**Table 1  Information on ground-level PM$_{2.5}$ monitoring.**

| Monitoring sites code | City | Longitude (°E) | Latitude (°N) | Time span |
|---|---|---|---|---|
| 1490A | Urumqi | 87.5801 | 43.8303 | 20150101–20210229 |
| 1491A | Urumqi | 87.6046 | 43.768 | 20150101–20210229 |
| 1492A | Urumqi | 87.4754 | 43.9469 | 20150101–20201231 |
| 1493A | Urumqi | 87.5525 | 43.8711 | 20150101–20210229 |
| 1494A | Urumqi | 87.6432 | 43.831 | 20150101–20210229 |
| 1495A | Urumqi | 87.4171 | 43.8729 | 20150101–20170208 |
| 1496A | Urumqi | 87.6444 | 43.962 | 20150101–20201231 |
| 1951A | Karamay | 84.8861 | 45.6033 | 20150101–20210229 |
| 1952A | Karamay | 84.8897 | 45.5828 | 20150101–20201231 |
| 1953A | Karamay | 85.1186 | 45.6886 | 20150101–20210229 |
| 1954A | Karamay | 84.8983 | 44.3336 | 20150101–20210229 |
| 1955A | Karamay | 85.6931 | 46.0872 | 20150101–20210229 |
| 1956A | Korla | 86.1461 | 41.7511 | 20150101–20210229 |
| 1957A | Korla | 86.2022 | 41.7192 | 20150101–20210229 |
| 1958A | Korla | 86.2381 | 41.7128 | 20150101–20210229 |
| 2686A | Turpan | 89.191 | 42.9409 | 20150101–20210229 |
| 2687A | Turpan | 89.1673 | 42.9559 | 20150101–20210229 |
| 2688A | Hami | 93.5128 | 42.8172 | 20150101–20210229 |
| 2689A | Hami | 93.4961 | 42.8328 | 20150101–20210229 |
| 2690A | Changji | 87.9897 | 44.1564 | 20150101–20210229 |
| 2691A | Changji | 87.2997 | 44.0114 | 20150101–20210229 |
| 2692A | Changji | 87.2717 | 44.0297 | 20150101–20210229 |
| 2693A | Bortala Mongol Autonomous Prefecture | 82.0485 | 44.9079 | 20150101–20210229 |
| 2694A | Bortala Mongol Autonomous Prefecture | 82.0806 | 44.8969 | 20150101–20201231 |
| 2695A | Aksu | 80.2828 | 41.1636 | 20150101–20210229 |
| 2696A | Aksu | 80.2956 | 41.1933 | 20150101–20210229 |
| 2697A | Kizilsu Kirghiz Autonomous Prefecture | 76.1861 | 39.7153 | 20150101–20210229 |
| 2698A | Kashgar | 75.9828 | 39.5371 | 20150101–20210229 |
| 2699A | Kashgar | 75.9771 | 39.4699 | 20150101–20210229 |
| 2700A | Kashgar | 75.9435 | 39.4365 | 20150101–20210229 |
| 2701A | Hetain | 79.9485 | 37.1152 | 20150101–20200623 |
| 2702A | Hetain | 79.9117 | 37.1013 | 20150101–20200620 |
| 2703A | Ili Kazak Autonomous Prefecture | 81.2815 | 43.9404 | 20150101–20201231 |
| 2704A | Ili Kazak Autonomous Prefecture | 81.2867 | 43.895 | 20150101–20210229 |
| 2705A | Ili Kazak Autonomous Prefecture | 81.3364 | 43.941 | 20150101–20201231 |
| 2706A | Tacheng | 82.9994 | 46.7432 | 20150101–20210229 |
| 2707A | Altay | 88.1214 | 47.9047 | 20150101–20210229 |
| 2708A | Altay | 88.1267 | 47.8515 | 20150101–20210229 |
| 2709A | Shihezi | 86.0497 | 44.2967 | 20150101–20210229 |
| 2710A | Shihezi | 86.0697 | 44.3075 | 20150101–20201231 |
| 2711A | Wujiaqu | 87.5475 | 44.1756 | 20150101–20210229 |
### *Random forest*

Random forest is a popular machine learning method that was proposed by Breiman (*Breiman, 2001*). This method has been used for prediction and categorization tasks in a variety of applications. The RF technique consists of a succession of computer-generated decision trees that can extract information from complex input data and learn the highly nonlinear relationship between input and goal variables. Specifically, RF randomly selects multiple attributes from each node's attribute set to form a subset and then establishes rules to determine the best attribute from this subset and finally predicts based on the average value of all leaf nodes among all trees.

This study aimed to construct an RF model for estimating the $PM_{2.5}$ concentrations across Xinjiang and for studying its spatial distribution and regional differences. Considering that the air quality situation is likely to vary slightly over a six-year period, based on the valid data we obtained, the average value of the data was taken every 8 days, and null values and outliers were excluded; the final number of valid samples obtained by all 41 stations in Xinjiang for each of the six years during 2015–2020 was (in order) 1,258, 1,276, 1,365, 1,360, 1,398, and 1,338. In addition, we used two indicators of the RF model to measure variable importance, namely, the percent increase in the mean squared error (%IncMSE) and the increase in node purity (IncNodePurity), to rank the importance of the factors. Finally, based on the six-year overall situation, we selected 9 factors, which were entered as predictors: AOD, DEM, NDVI, SI, SP, DLR, USR, WS, and RH. A RF model was established for the AOD–$PM_{2.5}$ relationship, as shown in Eq. (1). This model was implemented in the R3.6.3 language, and the modeling data involved in this paper are shown in Table 2.

The RF model constructed in this study can be abbreviated as:

$$PM2.5 = RF(AOD, DEM, NDVI, SI, SP, DLR, USR, WS, RH) \tag{1}$$

where AOD is the aerosol optical depth; DEM is the elevation; NDVI is the normalized difference vegetation index; SI is the drought index; SP is the surface pressure; DLR denotes downward longwave radiation; USR denotes upward shortwave radiation; WS is the wind speed; and RH denotes relative humidity.

To validate the model, 10-fold cross-validation (CV) was used in this work. The complete training data set was randomly divided into ten subgroups for each CV, nine of which were utilized as training subsets and the remaining one was employed as a validation subset. The average value was taken as the final accuracy of the RF model, and the Pearson correlation coefficient (R), $R^2$ and RMSE were calculated over the entire data set consisting of 10 test subsets to evaluate the correlations between the estimated and observed $PM_{2.5}$ concentrations.

## RESULTS

### Spatial distribution of AOD across Xinjiang
#### *Spatial distribution characteristics of AOD*

The applicability of MCD19A2 data has been verified in numerous studies (*Chen et al., 2021*; *Li et al., 2021b*; *Tao et al., 2019*), so verification is not performed here. The spatial

**Table 2  Summary of dataset used for modeling.**

| Data name | Data Source | Variables | Units | Resolution |
|---|---|---|---|---|
| AOD products | National Aeronautics and Space Administration (NASA) | Terra MODIS AOD products Aqua MODIS AOD products | | 1 km |
| PM$_{2.5}$ data | China's national air quality real-time release platform | PM$_{2.5}$ | ug/m$^3$ | |
| Meteorological data | National Centers for Environmental Prediction, (NCEP) | Maximum temperature, 2m | K | |
| | | Minimum temperature, 2m | K | |
| | | humidity | Kg/kg | |
| | | Maximum humidity, 2m | Kg/kg | |
| | | Minimum humidity, 2m | Kg/kg | |
| | | Potential Evaporation Rate surface | W/m$^2$ | 0.2 arc degrees |
| | | Precipitation | Kg/m$^2$/s$^1$ | |
| | | Pressure surface | Pa | |
| | | Upward Long-Wave Radp Flux | W/m$^2$ | |
| | | Downward Long-Wave Radp Flux | W/m$^2$ | |
| | | Upward Short-Wave Radp Flux | W/m$^2$ | |
| | | Downward Short-Wave Radp Flux | W/m$^2$ | |
| | | temperature | K | |
| Auxiliary data | NASA LP DAAC at the USGS EROS center | LST_Day | K | 1 km |
| | NASA GES DISC at NASA Goddard Space Flight Center | wind | M/s | 0.1arc degrees |
| | NASA | NDVI | | 1 km |
| | NASA | EVI | | 1 km |
| | NCEP | DEM | gpm | 0.2 arc degrees |
| | Institute of Industrial Science, The University of Tokyo, Japan | SI | | 4 km |
| | Annual Statistical Bulletin | Population density | | |

distribution of AOD in Xinjiang from 2015 to 2020 is shown in Fig. 2. There is an important difference between the southern and northern parts of Xinjiang. The AOD in northern Xinjiang has remained at a low level (<0.2) for six years. This may be due to the extensive vegetation coverage and weaker sand and dust interference in northern Xinjiang than in southern Xinjiang, where the AOD has remained high for six years due to the contribution of dust particles from the Taklimakan Desert, with the AOD values in some areas during 2018 and 2020 exceeding 0.4. There was an AOD inflection point in 2018 (0.222). During the preceding three years (2015–2017), AOD gradually decreased, which may have been linked to the "coal to gas" policy that Xinjiang began to implement in 2012; in contrast, since 2018, AOD has been volatile, which may be related to the advancement of Xinjiang's economic development strategy (*Li et al., 2021b*).

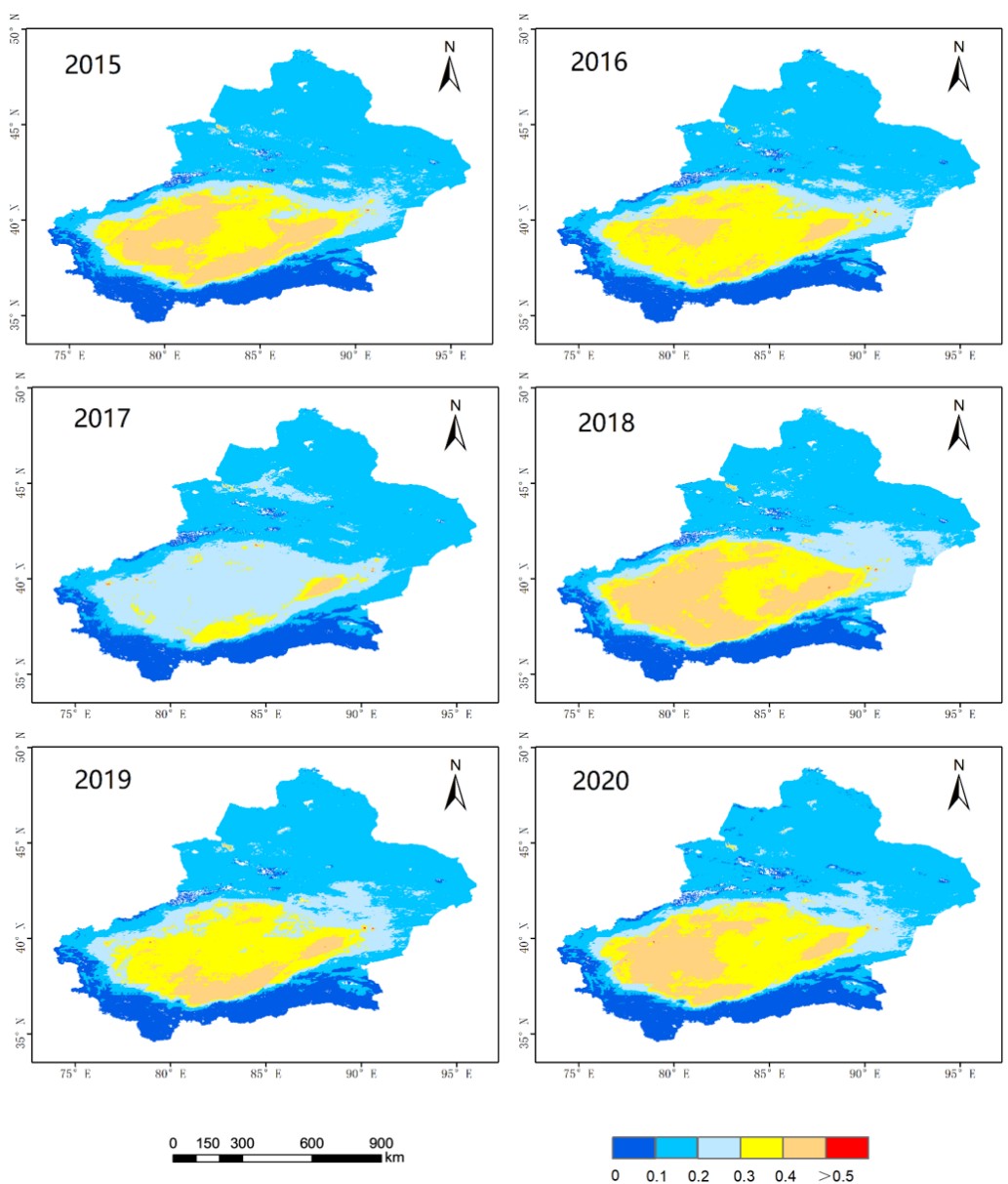

**Figure 2 Spatial distributions of AOD of 2015–2020.** The blue part represents a low value, and the red part represents a high value.

### Characteristics of the seasonal distribution of AOD

In this study, March, April, and May are grouped into spring, June, July, and August are grouped into summer, September, October, and November are grouped into autumn, and December and January and February are grouped into winter. Figure 3 shows the seasonally averaged spatial distributions of AOD from 2015 to 2020. The AOD magnitude decreased in the order of spring (0.342) > summer (0.219) > autumn (0.166) (AOD was not analyzed in the winter because of the difficulties in the inversion, the absence of AOD data, and the overall low AOD). Because spring brings more sand and dust, the area's overall AOD value

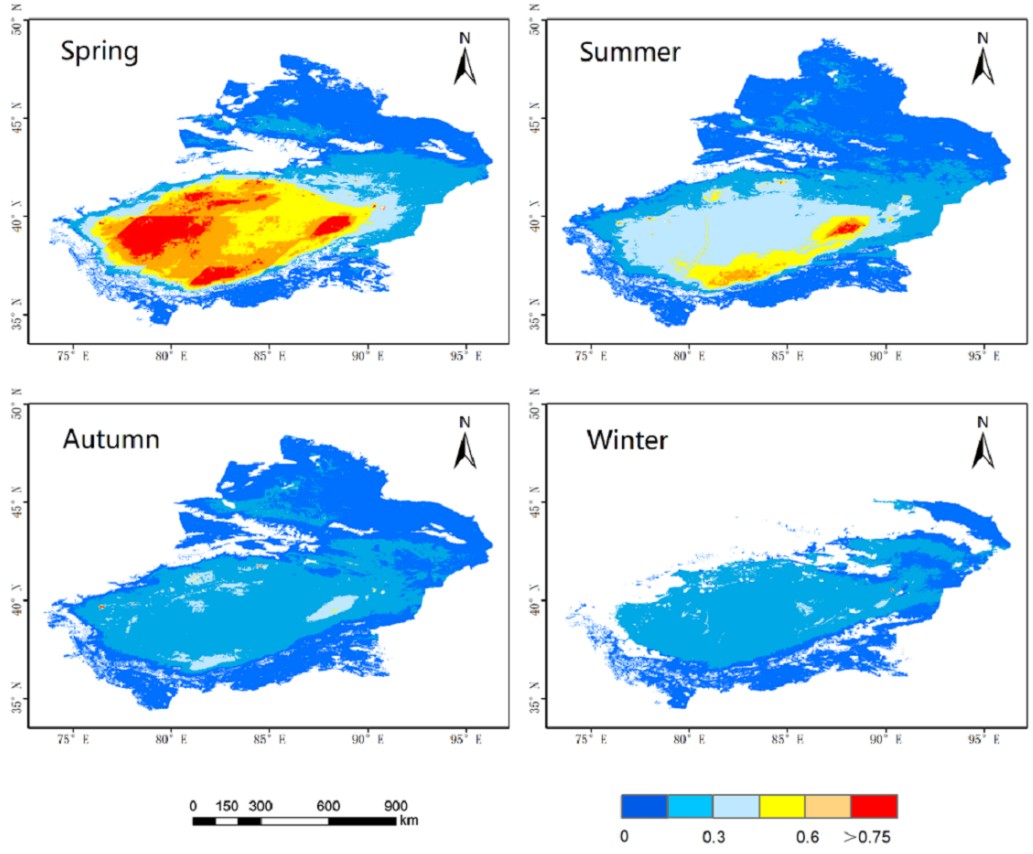

**Figure 3** **Seasonal mean AOD distribution.** The blue part represents a low value, and the red part represents a high value.

was higher than in the other seasons. In contrast, AOD was lower in summer; however, some regions in southern Xinjiang still exhibited high AOD values due to the influences of summer dust and storms. The climate was dry in autumn, with regular Arctic air intrusions and a stable air structure. Overall, the AOD has declined, indicating a state of diffusion (*Zhang et al., 2016*).

## Descriptive statistics

A summary of the ground-monitored $PM_{2.5}$ in Xinjiang during the study period is shown in Table 3. From 2015 to 2020, the average concentration of $PM_{2.5}$ monitored by all 41 air quality monitoring stations in the study area was 52.95 µg m$^{-3}$, ranging from 2 to 494.9 µg m$^{-3}$, and the mean annual $PM_{2.5}$ concentration among all 41 monitoring stations during these six years were (in order) 59.5 µg m$^{-3}$, 58.5 µg m$^{-3}$, 52.3 µg m$^{-3}$, 52.3 µg m$^{-3}$, 49.9 µg m$^{-3}$ and 45.2 µg m$^{-3}$. Accordingly, the average annual concentrations of $PM_{2.5}$ were much higher than the first-level annual limit of $PM_{2.5}$ (35 µg m$^{-3}$) set in China's Ambient Air Quality Standard (GB3095-2012). Seasonally, the $PM_{2.5}$ value was highest in winter (93.63 µg m$^{-3}$) and lowest in summer (28.60 µg m$^{-3}$). During the study period, the

**Table 3** A summary of ground monitoring PM$_{2.5}$ concentrations ($\mu$g m$^{-3}$) in Xinjiang, China during 2015–2020.

| Year | No. sites | No. samples | Minimum | Median | Maximum | Mean | Standard deviation |
|------|-----------|-------------|---------|--------|---------|------|--------------------|
| 2015 | 41 | 1,796 | 2 | 38.2 | 376.1 | 59.5 | 56.8 |
| 2016 | 41 | 1,784 | 5 | 34.7 | 478.1 | 58.5 | 59.2 |
| 2017 | 41 | 1,720 | 5.7 | 35.5 | 450.2 | 52.3 | 47.1 |
| 2018 | 41 | 1,697 | 4.2 | 33.4 | 494.9 | 52.3 | 51.9 |
| 2019 | 41 | 1,750 | 3 | 30.2 | 442.2 | 49.9 | 50.3 |
| 2020 | 41 | 1,653 | 3 | 27.4 | 436.5 | 45.2 | 49.8 |

seasonal PM$_{2.5}$ concentrations in Xinjiang decreased in the following order: winter (93.63 $\mu$g m$^{-3}$) >spring (56.02 $\mu$g m$^{-3}$) >autumn (42.40 $\mu$g m$^{-3}$) >summer (28.60 $\mu$g m$^{-3}$).

## Model fitting and validation

The main purpose of this work was to verify the ability of the RF model to estimate PM$_{2.5}$ in Xinjiang. Figure 4 shows a frequency scatterplot of the 10-fold CV results for the RF model and the bagging algorithm model from 2015 to 2020 with the R, R$^2$ and RMSE of both models. The R values of the RF model during all six years are (in order) 0.855, 0.879, 0.853, 0.900, 0.888, and 0.901, which are much higher than those of the bagging algorithm model at 0.796, 0.772, 0.747, 0.810, 0.810, and 0.789, respectively. These values indicate relatively strong agreement between the RF-estimated and station-measured PM$_{2.5}$ values. The corresponding R$^2$ values of the RF model are 0.731, 0.773, 0.728, 0.810, 0.788, and 0.813, which are similarly higher than those of the bagging algorithm model (0.633, 0.596, 0.558, 0.656, 0.657, and 0.622, respectively). Furthermore, the RMSEs of the RF model are lower than those of the bagging algorithm model, which indicates that the RF model is more stable. This comparison of the CV results of the two models demonstrates that the RF model is better than the bagging algorithm model in different aspects for the entire six-year period.

## Estimated spatiotemporal distribution of PM$_{2.5}$ across Xinjiang
### Annual spatiotemporal variations

According to Fig. 5, the concentration of PM$_{2.5}$ in the study area was higher in southern Xinjiang than in northern Xinjiang from 2015 to 2020, and high PM$_{2.5}$ values were concentrated in the Tarim Basin. At the same time, high concentrations extended outwards, affecting neighboring areas and establishing secondary high-value zones. The Taklimakan Desert, the largest desert in China, is located in the Tarim Basin. Correspondingly, studies have revealed that during high dust periods, wind can blow up dust on the ground and suspend it in the atmosphere. If no precipitation falls, this dust can persist for several days after the dust has passed, causing an increase in PM$_{2.5}$. As a result, the Tarim Basin has become a hotspot for high PM$_{2.5}$ concentrations (*Hui et al., 2010*). The estimated average PM$_{2.5}$ concentrations in Xinjiang during these six years were (in order) 71.69 $\mu$g m$^{-3}$, 73.19 $\mu$g m$^{-3}$, 56.68 $\mu$g m$^{-3}$, 66.35 $\mu$g m$^{-3}$, 60.30 $\mu$g m$^{-3}$, and 55.70 $\mu$g m$^{-3}$. The lowest and highest values were recorded in 2020 and 2016, respectively.

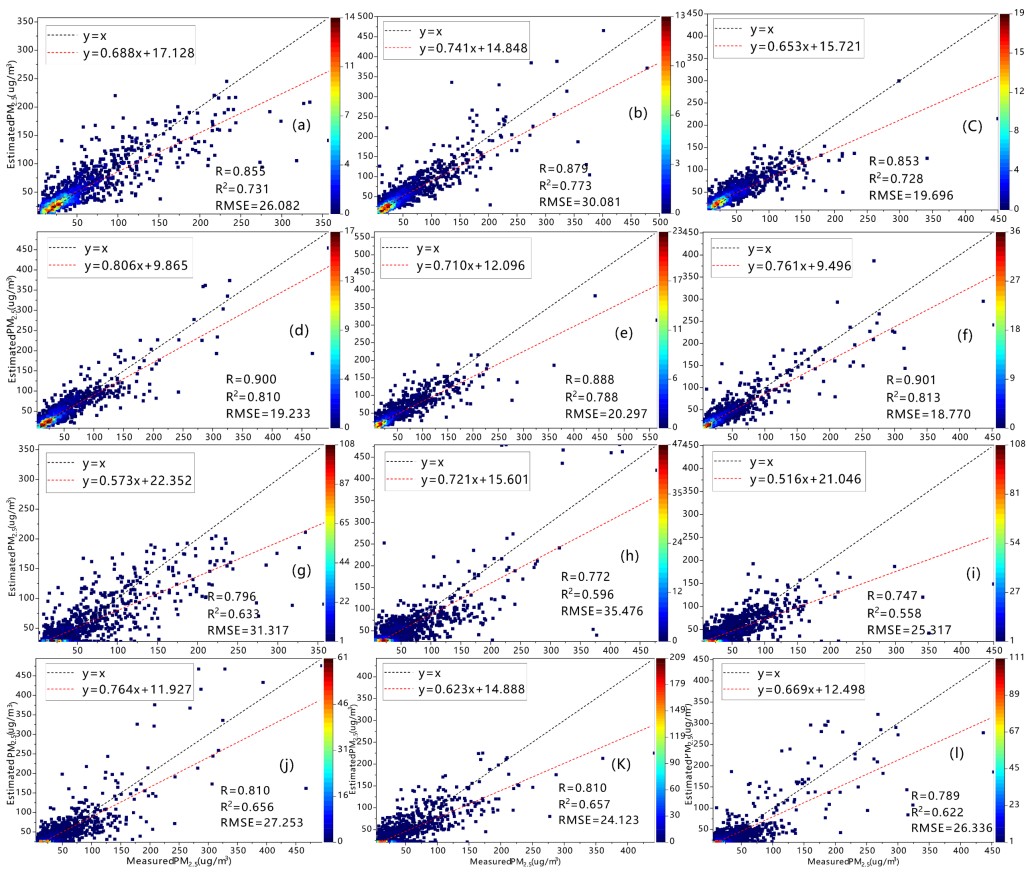

**Figure 4** Estimates and evaluations of predicted PM 2.5 concentrations based on cross-validation results of RF model (A–F) and Bagging Algorithm (G–L) 2015–2020 ($\times 10^{-4}$). The black line represents $y = X$, and the colored dots represent estimated and measured values.

In addition, the PM$_{2.5}$ concentrations in northern Xinjiang were lower than those in southern Xinjiang. Northern Xinjiang occupies a crucial strategic location: it is not only an important aspect of China's western development strategy but also Xinjiang's most prosperous region, as the vegetation coverage rate in northern Xinjiang is relatively high, which alleviates certain air pollution to a large extent. However, the population of northern Xinjiang is much denser than that of southern Xinjiang, which corresponds to more human activities in northern Xinjiang, that is, the increased burning of fossil fuels (coal, gasoline, diesel) and biomass (straw, firewood) and increased levels of building dust (*Wang et al., 2020*; *Wang et al., 2019*). Therefore, human activity is another source of PM$_{2.5}$ in northern Xinjiang.

### Seasonal spatiotemporal variations

Figure 6 shows the spatial distributions of the PM$_{2.5}$ concentration in Xinjiang from 2015 to 2020 estimated by the RF model. PM$_{2.5}$ decreased in the following order: winter (71.95 µg m$^{-3}$) > spring (64.76 µg m$^{-3}$) > autumn (46.01 µg m$^{-3}$) > summer (43.40 µg m$^{-3}$). The highest values in winter are attributed to the heating of homes throughout Xinjiang,

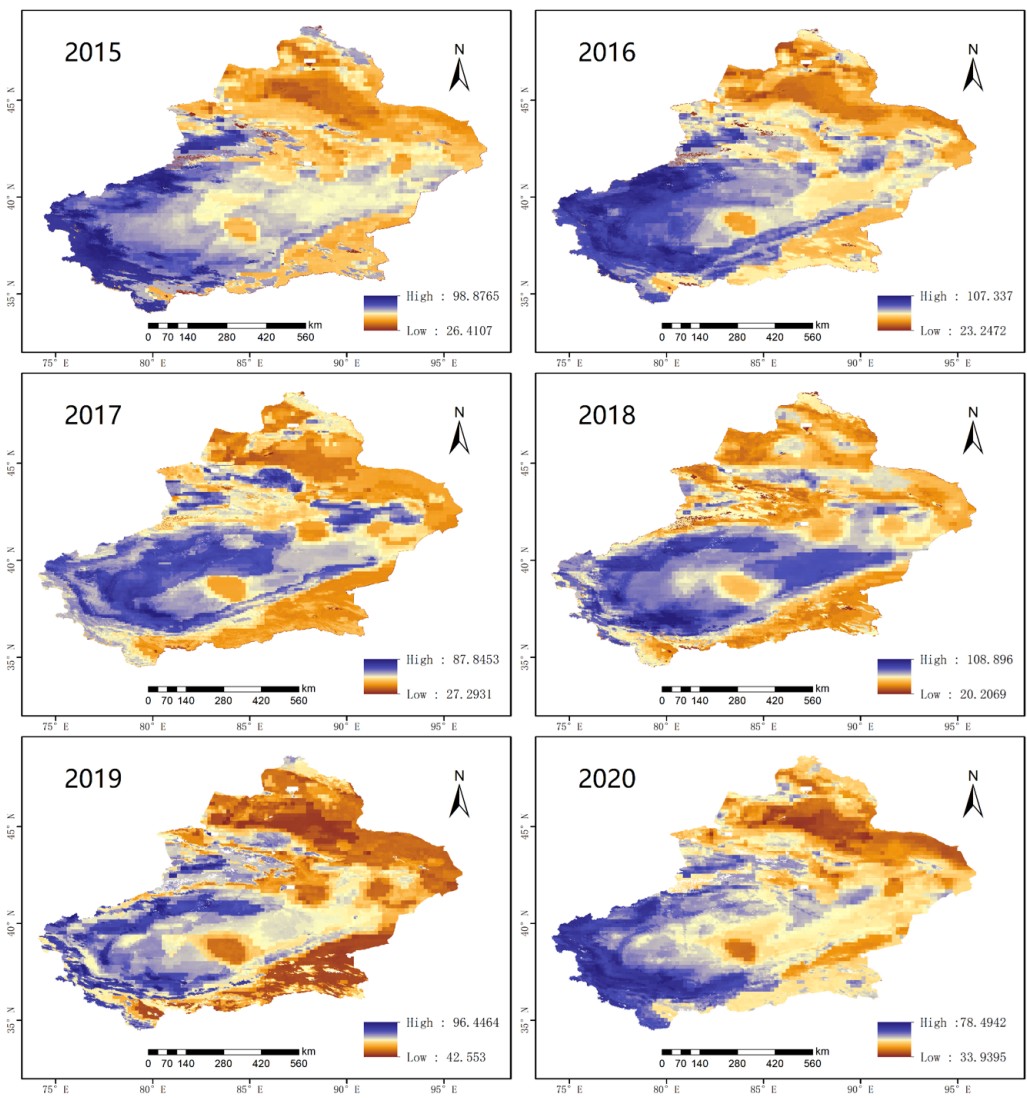

**Figure 5 The annual PM$_{2.5}$ concentration distributions in Xinjiang from 2015 to 2020.** Purple represents high values and yellow represents low values.

as heating in winter increases energy consumption and leads to higher concentrations of PM$_{2.5}$ (*Zhang et al., 2019a*). However, because the underlying surface is covered by snow, it is difficult for dust to be entrained into the atmosphere, and snow increases the surface albedo, which increases the difficulty of AOD inversion and even leads to data gaps in northern Xinjiang, resulting in the overall low PM$_{2.5}$ in winter mentioned above.

In spring, frequent sandstorms lead to higher PM$_{2.5}$ levels overall. When strong winds blow over the bare desert or bare soil, it causes wind erosion and entrains dust, aggravating air pollution in the surrounding area (*Chai et al., 2017*; *Liu et al., 2020*). In spring, the PM$_{2.5}$ concentrations in most areas of Xinjiang were greater than 56.64 μg m$^{-3}$. The Tarim Basin in southern Xinjiang has always been the main source of dust aerosols, but due

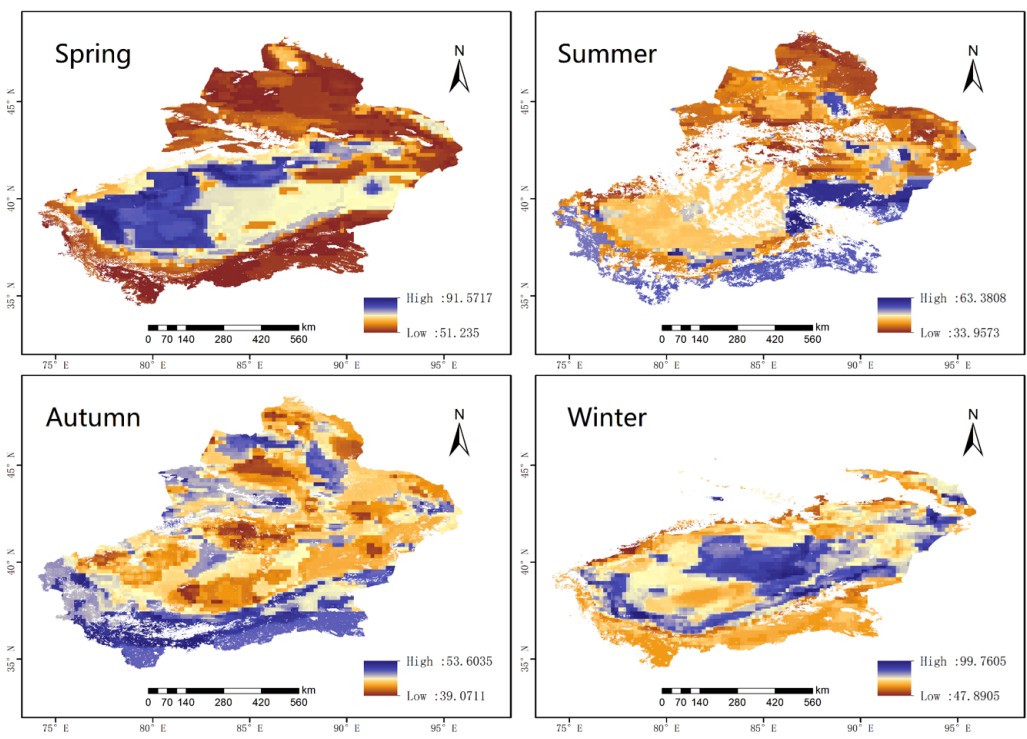

**Figure 6** **The predicted PM$_{2.5}$ concentrations from 2015 to 2020 in the four seasons based on satellite data.** Purple represents high values and yellow represents low values.

to topographical reasons, the internal wind speed is relatively low, and thus, horizontal diffusion is relatively weak, which concentrates PM$_{2.5}$ therein.

Summer was the season with the lowest PM$_{2.5}$ concentration. Because the surface air temperature in summer is higher, the atmospheric transparency is better, and the surface vegetation coverage is considerably enhanced. In addition, precipitation in summer is greater than that in other seasons, and rainfall has an obvious ability to remove dust from the atmosphere (*Zhang, Ding & Wang, 2017b*). Therefore, the PM$_{2.5}$ concentrations in summer were lower than those in the other seasons.

Among the four seasons, the average PM$_{2.5}$ value in autumn was the second-lowest. The low concentrations in autumn differ from those in summer because the atmosphere in the research area is characterized by locally high pressures, and the atmospheric system is rather stable in October (*Mogo et al., 2017*). As shown in Fig. 6, the PM$_{2.5}$ levels in most areas of Xinjiang remained low in autumn.

## DISCUSSION

The RF model adopted in this study is suitable for solving the complex nonlinear relationship between AOD and PM$_{2.5}$, which has been tested in previous studies. Our paper considers the unique weather conditions in Xinjiang, as well as a number of topographical and meteorological factors that are taken as independent input variables (*Jiang et al.,*

*2017*; *Yun et al., 2019*). Research shows that the prediction accuracy of an RF model is comparable to that of the algorithms presented in previous studies (*Li et al., 2019*; *Zhao et al., 2019*). For instance, *Li et al. (2021a)* combined RF machine learning with a generalized additive mixture model and incorporated visibility data and other meteorological factors to estimate the $PM_{2.5}$ concentrations in Iraq and Kuwait from 2001 to 2018, and their CV yielded an R of 0.71. In addition, our correlation is somewhat higher than that of *Xu et al. (2021a)* who used the Himawari Advanced Imager (AHI) AOD and other influencing factors to calculate the correlation between the $PM_{2.5}$ and AOD in 14 urban agglomerations, yielding an R range of 0.03~0.47. According to our findings, the RF model obtains a high correlation and is superior to most statistical regression models, including multiple linear regression (MLR) models (*Ji et al., 2019*) and generalized additive models (GAMs) (*Chen et al., 2018*). Additionally, the RF model is more accurate than some geostationary and combined models, such as the GWR model (*Bai et al., 2016*; *Zhai et al., 2018*), time-weighted regression (TWR) model (*Bai et al., 2016*), two-stage model (*He & Huang, 2018*), and principal component analysis GWR (PCA–GWR) model (*Zhai et al., 2018*). Therefore, the RF model in this study can better explain the spatial and temporal distributions of $PM_{2.5}$ in Xinjiang.

To develop a 1-km $PM_{2.5}$ surface map of Xinjiang, the MCD19A2 aerosol product was utilized. The data in this product are of higher quality and resolution than data with a 3-km, 10-km or even coarser resolution, which helps the RF model to serve better predictive functions (*Munchak et al., 2013*; *Yang, Xu & Jin, 2019*). AOD–$PM_{2.5}$ relationships are commonly used in environmental monitoring to estimate regional and global $PM_{2.5}$ patterns over time and space. Thus, high-resolution $PM_{2.5}$ concentrations are often retrieved from the archived data to study the health effects of long-term and short-term exposure to particles and to better monitor air quality in the future.

We expect that our research will have some impact on public health. Previous studies have focused mainly on coastal regions or areas with a concentrated monitoring system (*Lu et al., 2021*; *Pang et al., 2018*). In contrast, few studies have investigated Xinjiang, the central province of the Belt and Road Initiative and the main source of dust in China. Air pollution has occurred in Xinjiang as a result of human and natural factors acting together. Consequently, monitoring stations were gradually established across Xinjiang beginning in 2012. However, considering the vast, unpopulated area and the uneven distribution of stations, data from these stations are lacking, and continuity is an issue. In this study, a RF model was developed to invert the $PM_{2.5}$ concentration in Xinjiang and obtain its spatial distribution based on auxiliary data of meteorological and geographical variables. The results of this study will make it possible to assess exposure to air pollution in areas lacking an extensive array of monitoring stations or historical data of $PM_{2.5}$ concentrations. Furthermore, based on global data, our model can also be used to predict $PM_{2.5}$ concentrations in other parts of the world using predictors from those data sets.

At present, air pollution not only threatens human health but also restricts the sustainable development strategy of Xinjiang. Therefore, determining how to take concrete steps to prevent further deterioration due to pollution is an important task. In 2013, the Air Pollution Prevention and Control Action Plan was issued, and Xinjiang has since taken

measures to replace coal with gas and to retrofit or close enterprises with excessive emissions; moreover, the good days in Xinjiang each year have been summarized and reported. These actions all represent the government's contributions to curb air pollution from a policy perspective. Nevertheless, we must not only rely on government decision-making but also develop a better understanding of environmental protection, vigorously respond to national calls to action, follow the country's applicable rules, and do what we can on an individual level, such as campaigning to reduce carbon emissions. Traveling, volunteering to help plant public trees, refraining from using pyrotechnics and firecrackers, and other small actions can help reduce pollution and improve air quality.

## CONCLUSIONS

The inversion of $PM_{2.5}$ concentrations based on the RF model achieved good model performance, yielding R values for the six years during 2015–2020 of (in order) 0.855, 0.879, 0.853, 0.900, 0.888, and 0.901 and corresponding $R^2$ values of 0.731, 0.773, 0.728, 0.810, 0.788, and 0.813. Both metrics of the RF model are higher than those of the bagging algorithm model, and the RMSE of the RF model is lower than that of the bagging algorithm model; all of these outcomes show that the RF model performs well. Based on long time series of satellite data, the RF model can be used to reconstruct the spatial distribution of $PM_{2.5}$ at large spatiotemporal scales.

Over the past six years, the $PM_{2.5}$ concentrations were high in southern Xinjiang and low in northern Xinjiang. High values were concentrated mainly in the Tarim Basin, but they also diffused outward to form a secondary high-value area. In contrast, the vegetation coverage in northern Xinjiang is relatively high, and $PM_{2.5}$ in most areas remains low throughout the year.

Finally, the $PM_{2.5}$ concentrations in Xinjiang showed significant seasonality with the seasonal average decreasing as follows: winter (71.95 $\mu g\ m^{-3}$) > spring (64.76 $\mu g\ m^{-3}$) > autumn (46.01 $\mu g\ m^{-3}$) > summer (43.40 $\mu g\ m^{-3}$). Air pollution is relatively stable in summer and autumn, whereas pollution is the most serious in spring and winter. However, the inversion accuracy was poor in winter due to a lack of data.

### Funding
This work was supported by the National Natural Science Foundation of China (No. 41961059 and No.41771470). The funders had no role in study design, data collection and analysis, decision to publish, or preparation of the manuscript.

### Grant Disclosures
The following grant information was disclosed by the authors:
National Natural Science Foundation of China: 41961059, 41771470.

### Competing Interests
The authors declare there are no competing interests.
## Author Contributions

- XiaoYe Jin and Xiangyu Ge conceived and designed the experiments, performed the experiments, analyzed the data, prepared figures and/or tables, authored or reviewed drafts of the paper, and approved the final draft.
- Jianli Ding conceived and designed the experiments, prepared figures and/or tables, authored or reviewed drafts of the paper, and approved the final draft.
- Jie Liu conceived and designed the experiments, analyzed the data, prepared figures and/or tables, authored or reviewed drafts of the paper, and approved the final draft.
- Boqiang Xie, Shuang Zhao and Qiaozhen Zhao analyzed the data, prepared figures and/or tables, and approved the final draft.

## Data Availability

The original data and codes are available in the Supplementary Files.

## Supplemental Information

Supplemental information for this article can be found online at http://dx.doi.org/10.7717/peerj.13203#supplemental-information.

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
