# Peer review of "Machine learning driven by environmental covariates to estimate high-resolution PM2.5 in data-poor regions"

_PeerJ, doi:10.7717/peerj.13203_

## Round 0.1 · original submission · Major Revisions

Based on the comments from the two anonymous referees, major revisions are required to further improve your manuscript, particularly in the method description, result comparison and validation, and reference citation. Thank you for your submission to PeerJ.

Reviewer 1 ·

Basic reporting

no comment

Experimental design

In the section of model development,the paper take PM2.5 as response variable, aerosol optical ickness, annual precipitation, annual average temperature,183 wind speed, wind direction, relative humidity, urface temperature, surface pressure, the potential evaporation rate, DEM, normalized difference vegetation index and enhanced vegetation index, population density as predictor input variables.there are so many predictor input variables ,Are all these predictor variables necessary?For a prediction model, the fewer model parameters, the better the model。Therefore, it is suggested that before inputting these parameters into the random forest, first make a significance test on these parameters and select the variables with significant influence as the input parameters.

Validity of the findings

no comment

Reviewer 2 ·

Basic reporting

In this study, the authors present the estimation results of PM2.5 concentrations in the Xinjiang province of China. The article is well written, but it should be further improved to meet the standards of publication. Please refer to the specific comments listed below.

1. Relevant prior literature on spatio-temporal data-driven machine learning methods should be appropriately referenced. The authors only mention spatio-temporal data-driven machine learning in the title, whereas the comparison of different methods for utilizing spatio-temporal information is untouched in the article. The following papers are listed for reference.

[1] Wei, J., Huang, W., Li, Z., Xue, W., Peng, Y., Sun, L., Cribb, M., 2019. Estimating 1–km- resolution pm2.5 concentrations across China using the space-time random forest approach. Rem. Sens. Environ. 231, 111221.
[2] Yang, N., et al.2022. "Geographical and temporal encoding for improving the estimation of PM 2.5 concentrations in China using end-to-end gradient boosting.". Rem. Sens. Environ.

2. In Figure 1, the legend of the figure is not labeled as English words. In all figures of this article, it is confused for readers that all the high values are marked as blue and the low values are marked as red, which are opposite to the general studies. The authors should explain why they adopted this type of figure legend.

Experimental design

1. I thank the authors for providing the raw data. The current raw data only contains the PM2.5 data, whereas the AOD data and the meteorological data of training samples are not presented in the file of raw data. Readers cannot reproduce the experiments via the current data file.

2. The code of cross-validation could not be found in the source code. Sufficient details of model hyperparameters are not described to replicate the experiments.

Validity of the findings

1. The article should clearly define the research question and show how the problems are solved. In lines 21-22, the authors claim that “We hypothesize that combining temporal and spatial information can improve the PM2.5 estimation accuracy because the spatiotemporal data expand the training sample of the model.” I suggest that authors should present the estimating results with and without temporal and spatial information to prove their hypothesis.

2. In this research, the model is not compared with other results of previous research in this study area. The aerosol emissions could change each year, and different number of training samples also affects the model performance. The authors need to add some experiments and discussions about the model comparison.

3. Please provide the standard deviations based on the 10-fold results to show whether the proposed model is robust are statistically significant.

---

## Round 0.2 · Minor Revisions

Your revised manuscript has been significantly improved. The reviewer suggests accepting it after further polishing the language. Please ask a professional English editor to help to polish the language, and then re-submit the manuscript as soon as possible, better within one week. Thanks.

Reviewer 2 ·

Basic reporting

This article needs to be polished further with professional English proofreading.

Experimental design

no comment

Validity of the findings

no comment

Additional comments

no comment

---

## Round 0.3 · accepted · Accept

The language of the revised manuscript has been significantly improved. It can be accepted for publication in the journal of PeerJ. Expect your future work to submit to PeerJ.